# Recent Advances in Molecular Mechanism and Breeding Utilization of Brown Planthopper Resistance Genes in Rice: An Integrated Review

**DOI:** 10.3390/ijms241512061

**Published:** 2023-07-27

**Authors:** Liuhui Yan, Tongping Luo, Dahui Huang, Minyi Wei, Zengfeng Ma, Chi Liu, Yuanyuan Qin, Xiaolong Zhou, Yingping Lu, Rongbai Li, Gang Qin, Yuexiong Zhang

**Affiliations:** 1Guangxi Key Laboratory of Rice Genetics and Breeding, Rice Research Institute, Guangxi Academy of Agricultural Sciences, Nanning 530007, China; yanliuhui1995@163.com (L.Y.); ttp168128@sina.com (T.L.); hdh1103@163.com (D.H.); m18275845028@163.com (M.W.); 13737942181@163.com (Z.M.); liu_chi66@126.com (C.L.); tlldv7735283@163.com (X.Z.); 2Liuzhou Branch, Guangxi Academy of Agricultural Sciences, Liuzhou Research Center of Agricultural Sciences, Liuzhou 545000, China; m17671620451@163.com; 3State Key Laboratory for Conservation and Utilization of Subtropical Agro-Bioresources, College of Agriculture, Guangxi University, Nanning 530004, China; lirongbai@126.com; 4Agricultural Science and Technology Information Research Institute, Guangxi Academy of Agricultural Sciences, Nanning 530007, China; qyy3931@163.com

**Keywords:** rice, brown planthopper, resistance gene, molecular mechanism, breeding applications

## Abstract

Over half of the world’s population relies on rice as their staple food. The brown planthopper (*Nilaparvata lugens* Stål, BPH) is a significant insect pest that leads to global reductions in rice yields. Breeding rice varieties that are resistant to BPH has been acknowledged as the most cost-effective and efficient strategy to mitigate BPH infestation. Consequently, the exploration of BPH-resistant genes in rice and the development of resistant rice varieties have become focal points of interest and research for breeders. In this review, we summarized the latest advancements in the localization, cloning, molecular mechanisms, and breeding of BPH-resistant rice. Currently, a total of 70 BPH-resistant gene loci have been identified in rice, 64 out of 70 genes/QTLs were mapped on chromosomes 1, 2, 3, 4, 6, 8, 10, 11, and 12, respectively, with 17 of them successfully cloned. These genes primarily encode five types of proteins: lectin receptor kinase (LecRK), coiled-coil-nucleotide-binding-leucine-rich repeat (CC-NB-LRR), B3-DNA binding domain, leucine-rich repeat domain (LRD), and short consensus repeat (SCR). Through mediating plant hormone signaling, calcium ion signaling, protein kinase cascade activation of cell proliferation, transcription factors, and miRNA signaling pathways, these genes induce the deposition of callose and cell wall thickening in rice tissues, ultimately leading to the inhibition of BPH feeding and the formation of resistance mechanisms against BPH damage. Furthermore, we discussed the applications of these resistance genes in the genetic improvement and breeding of rice. Functional studies of these insect-resistant genes and the elucidation of their network mechanisms establish a strong theoretical foundation for investigating the interaction between rice and BPH. Furthermore, they provide ample genetic resources and technical support for achieving sustainable BPH control and developing innovative insect resistance strategies.

## 1. Introduction

Rice (*Oryza sativa* L.) is a globally significant staple food crop, serving as the primary source of sustenance for over half of the world’s population [1,2,3]. The brown planthopper (Hemiptera: Delphacidae) is a major pest of rice and one of the primary biotic factors causing yield reduction [4,5]. BPH directly damages rice plants by feeding on their tissues through piercing-sucking mouthparts. Additionally, BPH acts as a vector for various plant pathogens, transmitting viral diseases, such as grassy stunt virus (RGSV) and ragged stunt virus (RRSV), which indirectly harm rice crops [6,7,8]. In modern agricultural practices, chemical insecticides are the primary means of controlling BPH infestation [3,9,10]. However, the excessive use of chemical insecticides can lead to issues such as pesticide residues, detrimental effects on human health, environmental pollution, disruption of field ecological balance, and increased production costs. Prolonged use of chemical insecticides can induce BPH to adapt to new environments, resulting in the emergence of more destructive biotypes with heightened virulence. Consequently, this leads to the development of insecticide resistance, creating a need for increased reliance on chemical insecticides and perpetuating a detrimental cycle [10,11,12,13]. BPH’s resistance to chemical insecticides renders chemical control ineffective for achieving sustainable management of the pest [13,14]. Hence, the exploration and utilization of intrinsic resistance genes in rice, with the objective of breeding and promoting BPH-resistant rice varieties, are widely recognized as the most economically effective measures for achieving sustainable pest control [14,15,16]. The concept of “Green Super Rice”, proposed in recent years, emphasizes the importance of insect resistance as one of the essential traits [14,17]. Hence, the exploration and utilization of BPH-resistant genes in rice hold immense importance for implementing biological pest control in rice production.

In this review, we provided a comprehensive overview of the recent progress in the field of BPH-resistant genes in rice. Specifically, we highlighted the cloning of these genes and elucidated the molecular regulatory networks that exist among them. Additionally, we addressed the challenges associated with naming BPH-resistant genes in rice and discussed their application in rice breeding. The insights gained from this review contributed significantly to the development of sustainable biological control strategies for managing BPH infestations.

## 2. BPH-Resistant Rice Germplasm Resources

Since the 1960s, scientists have been systematically screening and identifying a large number of BPH-resistant materials from wild rice and cultivated rice varieties in Southeast Asia [18]. Currently, the identified 70 BPH-resistance genes/quantitative trait loci (QTLs) (50 genes and 20 QTLs) are equally derived from wild rice and *O. sativa* (Figure 1, Table 1). Of which 14, 10, 4, 3, 2, 1, and 1 resistance genes/QTLs or QTLs were derived from *O. rufipogon*, *O. officinalis*, *O.*
*nivara*, *O. minuta*, *O. australiensis*, *O. eichingeri, and O. latifolia*, respectively, and the remaining 35 resistance genes/QTLs were from *O. sativa*. The exploration and identification of these resistant germplasm resources provide a diverse gene pool for the breeding and genetic improvement of BPH-resistant rice varieties [18,19].

## 3. Mapping of BPH-Resistant Genes/QTLs in Rice

In 1971, the first BPH-resistant gene, *Bph1*, was identified and located by the International Rice Research Institute (IRRI). To date, a total of 70 BPH-resistant genes/QTLs (50 genes and 20 QTLs) have been identified in rice. They are mainly distributed on chromosomes 3, 4, 6, and 12 (Figure 2, Table 1). To avoid confusion caused by duplicate gene names associated with BPH resistance, this study implemented a ranking system based on the chronological order of gene discovery. Among the localized BPH-resistant genes in rice, most genes are clustered on chromosomes 3, 4, 6, and 12. Ten genes [*Bph1*, *Bph2*, *Bph7*, *Bph9*, *Bph10*, *qBph12*, *Bph18*, *Bph19*(*t*)*-2*, *Bph21*(*t*)*-1*, and *Bph26*] are clustered between the molecular markers RM7102 and RM17 on chromosome 12. Moreover, 30 genes are clustered in three regions of chromosome 4 [*Bph30* and *Bph41*-1 located in the 0.90–1.10 Mb segment between molecular markers SSR28 and SWRm_01522; *Bph40-2*, *Bph41-2*, *Bph12-1*, *qBph4-1*, *Bph35*, *Bph12*(*t*), *Bph36*, *qBph4.2*, *qBph4-2*, *qBph4.3*, *Bph15*, *Bph17*, *Bph3-2*, *qBph4.4*, *Bph20*(*t*)*-1*, *Bph42-2*, *Bph45* and *Bph38* located between molecular markers W4_4_3 and YM190; *Bph27*, *Bph22*(*t*)*-2*, *Bph33*, *Bph16*, *Bph12-2*, *Bph42-1*, *Bph27*(*t*), *Bph34*, *Bph44*, *Bph6,* and *Bph18*(*t*) located between molecular markers RM16846 and RM6506]. Six genes [*Bph11*(*t*), *qBph3*, *Bph13*(*t*)*-2*, *Bph19*(*t*)*-1*, *Bph31*, and *Bph14*] are located on chromosome 3. *Bph3-1*, *Bph4*, *Bph20*(*t*)*-2*, *Bph29*, *qBph6*, *Bph32*, *Bph37-2,* and *Bph25* are located on chromosome 6. Additionally, *Bph37-1*, *Bph38*(*t*), *Bph13*(*t*)*-1*, *Bph23*(*t*)*-2*, *qbph8*, *Bph21*(*t*)*-2*, *Bph28*(*t*), *Bph43,* and *qbph11* are located on chromosomes 1, 2, 8, 10, and 11, respectively. These clustered genes may represent different genes closely linked, different alleles at the same locus, or the same allele responding differently to different biotypes of BPH.

## 4. Cloning of BPH-Resistant Genes in Rice

So far, a total of 17 genes associated with resistance to BPH in rice have been cloned, of which eight BPH-resistance genes (*Bph1*, *Bph2*, *Bph7*, *Bph9*, *Bph10*, *Bph18*, *Bph21*, and *Bph26*) are multiple alleles of the same locus. These genes primarily encode five types of proteins: lectin receptor kinase (LRK), coiled-coil-nucleotide binding-leucine rich repeat (CC-NB-LRR), B3-DNA binding domain, leucine-rich repeat domain (LRD), and short consensus repeat (SCR) (Table 2).

### 4.1. Genes Associated with LecRKs

*Bph3* is a broad-spectrum and durable BPH-resistant gene derived from Rathu Heenati (RH) variety. It confers high resistance to four types of BPH biotypes as well as the white-backed planthopper. Initially, *Bph3* was mapped between the markers RM588 and RM589 on chromosome 6, while Liu et al. (2015) have later refined its location to the short arm of chromosome 4 and successfully cloned it [24,25,42]. *Bph3* is a gene cluster composed of three genes encoding plant LecRKs (*OsLecRKs*) [25]. Individual or combined transfer of one or two genes into susceptible varieties results in partial improvement of resistance to BPH, while the simultaneous transfer of all three genes further enhances the resistance [25]. Using RNAi technology to downregulate the expression of *OsLecRKs* in resistant parental lines significantly reduces their BPH resistance. Furthermore, through the identification of BPH resistance in recombinants selected from genetic populations, it is found that the absence of a single gene within the gene cluster significantly reduces resistance to BPH. These findings indicate that the broad-spectrum and durable resistance of *Bph3* is collectively controlled by the *OsLecRK* gene cluster, and each member of the gene cluster contributes additively to BPH resistance. *Bph15*, located on the short arm of chromosome 4 in the introgression line B5 derived from wild medicinal rice, is another BPH-resistant gene. It also contains a clustered region of *OsLecRK* genes [41,84,85]. Sequence analysis reveals that the amino acid sequences of the three *OsLecRK* genes within the *Bph15* region are identical to those of *Bph3*, suggesting that they may represent the same gene [83].

### 4.2. Genes Associated with CC-NB-LRR Proteins

*Bph14* is the first cloned BPH-resistant gene isolated from the medicinal wild rice material B5. It encodes a protein with coiled-coil (CC), nucleotide-binding (NB), and leucine-rich repeat (LRR) domains, known as CC-NB-LRR protein [40]. The BPH resistance levels conferred by the CC and NB domains alone, as well as their combination, are comparable to those exhibited by the full-length (FL) *Bph14* protein [86]. Both domains are capable of activating the SA signaling pathway and defense gene expression. Moreover, the resistance domains form a homologous complex with the FL *Bph14* protein, which interacts with transcription factors (TFs) *WRKY46* and *WRKY72* [86]. In rice protoplasts, the expression of FL *Bph14* or its CC, NB, and CN domains increases the accumulation of *WRKY46* and *WRKY72*, thereby enhancing the transactivation activity dependent on *WRKY46* and *WRKY*72 [86]. *WRKY46* and *WRKY72* can bind to the promoters of the receptor-like cytoplasmic kinase gene *RLCK281* and the callose synthase gene *LOC_Os01g67364*.1 [86]. Additionally, the expression of the *Bph14* gene is upregulated after BPH feeding, activating the salicylic acid (SA) signaling pathway. Furthermore, BPH feeding induces the accumulation of callose, leading to sieve tube blockage, which reduces the insect’s feeding, growth, and survival [86]. This mechanism is considered an important defense mechanism in rice against BPH.

In addition to *Bph14*, other genes such as *Bph9* and its allelic genes (*Bph1*, *Bph2*, *Bph7*, *Bph10*, *Bph18*, *Bph21*, and *Bph26*) also encode CC-NB-LRR proteins. However, *Bph9* and its allelic genes encode proteins with two NBS domains (CC-NB-NB-LRR, CNNL) [33]. Sequence analysis has revealed significant differences between the two NBS domains of *Bph9*. NBS2 possesses a complete NBS structure, while NBS1 lacks the sequence corresponding to ARC2 [32]. These sequence differences confer distinct functions to the two domains. NBS2 acts as a molecular switch, inhibiting the activity of the CC domain and subsequently relieving the inhibition of the LRR domain after the interaction. Due to the lack of the sequence corresponding to ARC2, NBS1 does not possess the ability to inhibit the activity of the CC domain. However, the *Bph9*-mediated resistance relies on the activation of downstream hypersensitive response (HR) by the amino acids at positions 97-115 in the CC domain [32]. In the *Bph9*-mediated BPH resistance, the LRR domain plays the role of a signal recognizer, while the two NBS domains act as regulators, and the CC domain carries out the activation of downstream immune responses. *Bph9* was initially located on the long arm of chromosome 12 along with *Bph1*, *Bph2*, *Bph7*, *Bph10*, *Bph18*, *Bph21,* and *Bph26* [20,21,22,23,30,32,33,34,44,45,47,52]. Among them, *Bph2* is identical to *Bph26* [52], *Bph9* shares 91.32% and 96.05% nucleotide sequence identity with *Bph26* and *Bph18*, respectively. The nucleotide and deduced amino acid sequences in rice varieties harboring *Bph1*, *Bph10*, and *Bph21* were identical to *Bph18*, whereas the sequence from T12 (harboring *Bph7*) indicated a distinct allele [20,30,34,45,47]. Together with their chromosome position and genomic sequence, these data confirmed that eight BPH-resistance genes (*Bph1*, *Bph2*, *Bph7*, *Bph9*, *Bph10*, *Bph18*, *Bph21*, and *Bph26*) are multiple alleles of the same locus [33]. Based on the sequence variations of these allelic genes, they can be classified into four allelic types: *Bph1/9-1* (*Bph1*, *Bph10*, *Bph18*, and *Bph21*), *Bph1/9-2* (*Bph2* and *Bph26*), *Bph1/9-7* (*Bph7*), and *Bph1/9-9* (*Bph9*) [33]. Interestingly, these four allelic genes exhibit different resistance spectra. *Bph1/9-1* confers resistance to biotypes I and III, *Bph1/9-2* confers resistance to biotypes I and II, while *Bph1/9-7* and *Bph1/9-9* confer resistance to biotypes I, II, and III simultaneously [33]. The study of allelic variations and different resistance spectra of *Bph9* provides important clues for elucidating the molecular mechanisms underlying rice’s response to BPH biotype variations. It also serves as a valuable reference for the exploration and utilization of different allelic genes associated with resistance. 

*Bph37* is another BPH-resistant gene encoding a CC-NB-LRR-like protein found in rice. What sets it apart is its encoding of a CC-NB protein that lacks the LRR domain. The absence of the LRR domain is a result of a 1-bp insertion in the second exon, which leads to the premature termination of protein translation [66]. The molecular mechanism responsible for *Bph37*-mediated insect resistance is still unknown, but the distinctive CC-NB structural arrangement suggests that the CC and NB domains may play separate roles in BPH resistance.

### 4.3. LRD Protein-Related Genes

*Bph6* is a broad-spectrum BPH-resistant gene derived from the cultivated rice variety Swarnalata, located on the long arm of chromosome 4 [87]. It encodes a novel protein found in the Exocyst complex. *Bph6* exhibits strong expression in the thick-walled tissues of rice leaf sheaths, leaf blades, vascular bundles, and companion cells, and this expression is not induced by BPH feeding [29]. Research suggests that the *BPH6*-*OsEXO70H3*-*OsSMASL* module enhances rice resistance to BPH by regulating the accumulation of cell wall lignin [88]. Zheng et al. (2020) have found that nymphs and adults of BPH feeding on NIL-*Bph6* rice plants exhibit reduced weight gain, and in some cases, negative growth, indicating suppressed feeding by the BPH [89]. Overall, *Bph6* plays a crucial role in maintaining and enhancing the development of thick-walled tissues and vascular bundles. It regulates the accumulation of cell wall lignin, which in turn inhibits BPH feeding, effectively impeding their growth and development. This multifaceted resistance mechanism is likely the main reason for its broad-spectrum BPH resistance. However, the specific role of *Bph6* in regulating plant hormone response pathways and BPH lipid metabolism, as well as how it interacts with different resistance mechanisms, remains unknown.

*Bph30* is a BPH-resistant gene located on the short arm of chromosome 4 in the AC1613 rice variety [57]. It encodes a protein that contains only two leucine-rich domains (LRDs) [57]. *Bph30* is localized to the endoplasmic reticulum, vacuolar membrane, and vesicles, and it shows high expression in thick-walled tissues [57]. It prevents BPH feeding by increasing the hardness and thickness of the cell walls in these tissues. This structural change makes it challenging for the insect’s mouthparts to penetrate the sclerenchyma, effectively hindering their ability to feed [57]. Sequence analysis of *Bph30* reveals that it belongs to a novel BPH-resistant gene family. Homologous gene analysis and whole-genome association studies have identified another BPH gene, *Bph40*, belonging to the same family in three rice materials, namely SE232, SE67, and C334 [57]. Its resistance mechanism is similar to that of *Bph30*. Furthermore, gene homology analysis has identified a total of 27 *Bph30*-like genes in rice. Apart from *Bph30* and *Bph40*, which have been proven to confer BPH resistance, it is yet to be determined if the remaining genes exhibit BPH resistance. Further research on the *Bph30* gene family may provide an effective pathway for discovering new BPH genes.

### 4.4. B3-DNA Domain Protein Gene

*Bph29* is a BPH-resistant gene isolated and cloned from chromosome 6 of the introgression line RBPH54, derived from wild rice (*Oryza rufipogon*) in Guangxi Province [56]. It encodes a nuclear-localized B3-DNA binding protein. *Bph29* shows specific expression in the vascular system of roots and leaves when BPHs feed on the plant. In response to BPH feeding, *Bph29* activates the SA signaling pathway and inhibits the jasmonic acid (JA)/ethylene (Et)-dependent pathway, similar to the defense response of plants against pathogens [56].

### 4.5. SCR Domain Protein-Related Gene

Ren et al. (2016) have conducted DNA sequencing and bioinformatics analysis on a 190-kb region between the markers RM19291 and RM8072 on the short arm of chromosome 6 in the broad-spectrum and durable BPH-resistant variety Ptb33. They have identified a BPH-resistant gene, *Bph32*, which encodes a membrane-localized protein with an SCR domain. Sequence comparison reveals that *Bph32* is identical to the allelic gene found in wild rice species. Analysis of *Bph32* expression patterns shows high expression in the leaf sheath, and it remains highly expressed at 2 h and 24 h after BPH feeding, suggesting that *Bph32* may counteract BPH infestation by inhibiting their feeding behavior [90].

## 5. Molecular Mechanism of Rice Resistance to BPH

The innate immune system of rice consists of a two-layer defense system comprising pattern-triggered immunity (PTI) and effector-triggered immunity (ETI), which are initiated by the recognition of BPH-associated molecular patterns (PAMPs or MAMPs) by cell membrane receptor kinases (such as *Bph3*) and effectors by intracellular proteins (such as *Bph6*, *Bph14*, and *Bph9*) respectively [91]. PTI or ETI activates a series of defense-related signaling pathways, including changes in calcium ion concentration, activation of MAPK pathways, protein phosphorylation, regulation of plant hormones, and expression of defense genes. These pathways induce the accumulation of callose in sieve tubes, leading to the blockage of sieve tubes, as well as the accumulation of cellulose, hemicellulose, and lignin in leaf sheath cell walls, resulting in cell wall thickening. These structural changes inhibit BPH feeding, growth, and reproduction, ultimately forming a resistance mechanism to counteract BPH infestation (Figure 3).

### 5.1. Recognition of Feeding Signals of Rice to BPH

After BPH feeding, its saliva is secreted into rice tissues. The saliva consists of various components with different functions. It can protect and lubricate the stylet, facilitating continuous feeding by the BPH. Some components of the saliva can be recognized by intracellular receptor proteins, activating defense responses to inhibit BPH feeding. Additionally, certain components can act to shield rice defense responses, promoting BPH feeding. NlShp encodes a salivary sheath protein that plays a crucial role in affecting saliva sheath formation and the normal feeding and survival of BPH on rice [93]. NlSEF1 encodes an EF-hand calcium-binding protein highly expressed in the BPH salivary gland. Further studies have shown that NlSEF1 protein binds to calcium, reducing calcium levels, which in turn lowers H_2_O_2_ levels and reduces callose accumulation, facilitating BPH feeding. Inhibition of NlSEF1 expression suppresses BPH feeding and reduces its survival rate [94]. Shangguan et al. (2018) have identified NlMLP (*N. lugens* mucin-like protein), a highly expressed mucin-like protein in the BPH salivary gland, using transcriptomic and proteomic analyses. Inhibition of NlMLP expression affects salivary sheath formation. Further research has revealed that the carboxyl terminal of NlMLP regulates Ca^2+^ flux and the JA signaling pathway, activating the expression of defense-related genes and inducing callose deposition [95]. Guo et al. (2023) identified and characterized the first insect salivary protein perceived by the plant immune receptor, which is a saliva protein BISP (BPH14-interacting salivary protein) of the brown planthopper (BPH) [96]. In susceptible varieties, BISP targets *OsRLCK185* and inhibits the basic defense. In varieties carrying the brown planthopper resistance gene *Bph14*, *Bph14* binds to BISP and activates the host immune response but inhibits rice growth [96]. BISP-*Bph14* binds to the selective autophagic cargo receptor *OsNBR1* and results in the degradation of BISP through the autophagic pathway, downregulating rice resistance against BPH and restoring the plant growth [96]. This study reveals the molecular mechanism underlying the balance of immunity and growth in the BISP-*BPH14*-*OsNBR1* interaction system by perceiving and regulating the protein level of insect effectors. β-glucosidase is one of the important components in BPH saliva and plays a key role in mediating rice defense responses. Treatment of mechanically damaged rice plants with β-glucosidase can activate rice signaling pathways similar to those induced by BPH infestation. The BPH salivary gland contains an endogenous β-1,4-glucanase, *NlEG1*, which can bypass rice defense responses mediated by JA and JA-Ile and degrade cell wall cellulose to facilitate BPH feeding. Downregulation of *NlEG1* inhibits BPH feeding, growth, and reproduction [95]. Furthermore, Kobayashi et al. (2014) have identified a recessive virulence gene, *vBph1*, which can overcome the resistance conferred by the BPH-resistant gene *Bph1* [97]. These studies provide important insights into the molecular mechanisms underlying the interaction between rice and BPH.

### 5.2. Signaling of Rice Defense against BPH

After perceiving the feeding signals of BPH, rice activates various signaling pathways, including plant hormones, calcium ion signaling, mitogen-activated protein kinase (MAPK) cascades, and TFs, to initiate appropriate defense responses against insect infestation.

#### 5.2.1. Plant Hormones

Plant hormones play a crucial role in regulating defense responses against BPH in rice. Hormones, such as salicylic acid (SA), jasmonic acid (JA), and gibberellins (GAs), have been implicated. Research indicates that SA positively regulates rice resistance to BPH, while JA exhibits a negative regulatory effect. BPH-resistant genes, such as *Bph14* and *Bph29*, activate the SA signaling pathway while suppressing the JA signaling pathway, thereby enhancing rice resistance to BPH [40,56]. Studies have also shown that the effects of SA and JA differ depending on the genetic background of resistance genes. For instance, the *OsHLH61* gene encoding a helix-loop-helix (HLH) protein is induced by BPH feeding and JA but is suppressed by SA. Further investigation has revealed that *OsHLH61* forms a dimer with *OsbHLH96*, leading to the modulation of pathogen-related (PR) genes and the crosstalk between SA and JA signaling pathways, thus mediating BPH resistance [98]. Additionally, it has been observed that both SA and JA signaling pathways can be simultaneously activated to positively regulate BPH resistance. In BPH-challenged near-isogenic lines of *Bph6* and *Bph9*, SA, JA, and JA-Ile are significantly upregulated. Compared to SA and JA, the involvement of GA in BPH resistance has been relatively less studied. *OsGID1*, a GA receptor gene, positively regulates rice resistance to BPH. Further research reveals that *OsGID1* enhances BPH resistance by modulating GA signaling to reduce JA and ethylene levels while promoting lignin accumulation [99]. The DELLA proteins, negative regulators of the GA pathway, also play a role. During the early defense response induced by BPH, the *OsSLR1* gene encoding a DELLA protein inhibits the GA pathway to activate defense-related signaling pathways in rice, leading to increased lignin and cellulose synthesis and enhanced resistance against BPH [100].

#### 5.2.2. Calcium Signaling

Calcium ions (Ca^2+^) are important secondary messengers in plant signal transduction pathways and play a crucial role in mediating various biological responses [101]. Ca^2+^ is a key component in the formation of callose [102]. The ability of the BPH-resistant gene *Bph14* to induce changes in Ca^2+^ levels, leading to callose deposition and inhibition of BPH feeding, is considered an important molecular mechanism underlying rice resistance to BPH [40]. NlSEF1, an EF-hand calcium-binding protein present in the salivary glands of BPHs, binds to Ca^2+^ and reduces its concentration, thereby weakening callose accumulation and facilitating BPH feeding [94].

#### 5.2.3. MAPK Cascade Reaction

MAPKs are a group of protein kinases that are activated by various cellular stimuli, including PAMPs, hormones, temperature changes, and other biotic and abiotic factors. MAPK cascades occur in all eukaryotic organisms and serve as a bridge to transmit different stimuli to downstream responsive genes, such as *MAPK3* and *MAPK6* [103]. In rice, several *OsMAPK* genes have been identified that activate defense-related gene expression and confer BPH resistance through the MAPK signaling cascade. For example, *Bphi008a* is induced by BPH feeding and ethylene, and it transmits the signal through the MAPK cascade to activate the expression of defense-related genes, thus conferring BPH resistance in rice. The MAPK cascade plays a crucial role in the signal transduction of genes related to BPH resistance [104].

#### 5.2.4. Transcription Factors

Rice defense responses are typically accompanied by the activation and expression of numerous defense genes, with TFs playing a crucial role in regulating the transcription of these defense-related genes [105]. TFs involved in rice’s response to biotic stress and resilience mainly include WRKY, MYB, NPR, ERF, and others [106]. *OsWRKY45* negatively regulates rice resistance against BPH [107,108]. *OsWRKY53* enhances rice resistance to BPH by increasing H_2_O_2_ levels and reducing the ethylene synthesis [109]. *MYB22* and *MYB30*, R2R3-type MYB TFs, positively regulate rice resistance to BPH [110,111]. The EAR motif of *MYB22* forms an MYB22-TOPLESS-HDAC1 complex, which negatively regulates the transcription of the *F3’H* (flavanone 3′-hydroxylase) gene, thereby enhancing rice resistance to BPH [110]. *OsMYB30* mediates rice resistance to BPH by regulating the expression of *OsPAL6* and *OsPAL8* genes [111]. *OsERF3*, through the MAPK and WRKY pathways, mediates the activation of JA, SA, ethylene, and H_2_O_2_ levels, thereby regulating early defense responses and conferring resistance to BPH in rice [112]. These studies illustrate the multifaceted roles of TFs in regulating rice defense responses.

### 5.3. miRNA-Mediated Rice BPH Resistance

MicroRNAs (miRNAs) play a crucial role in plant defense responses by specifically recognizing target mRNAs (TFs, signaling proteins, enzymes, etc.) and binding to them, leading to the degradation of target mRNAs, translation inhibition, or methylation-mediated negative regulation of gene expression [113]. Wu et al. (2017) [114], through the analysis of miRNA expression differences in BPH feeding on rice materials with or without the *Bph15* gene, have found that the miRNA changes are more significant in the *Bph15*-transgenic line after BPH feeding. Dai et al. (2019) have identified *OsmiR396*, which is associated with BPH resistance, through miRNA sequencing, revealing a novel resistance mechanism against BPH mediated by the *OsmiR396*-*OsGRF8*-*OsF3H*-flavonoid pathway [115]. *OsmiR156* negatively regulates rice resistance to BPH by increasing JA levels in rice [116]. These studies demonstrate the significant role of miRNAs in regulating rice resistance to BPH, and the identification of miRNAs involved in BPH resistance provides new insights for achieving BPH-resistant breeding strategies.

## 6. Marker-Assisted Breeding for BPH Resistance

Since the 1970s, IRRI has successfully developed and promoted several rice varieties resistant to the BPH by utilizing genes such as *Bph1*, *Bph2*, *Bph3*, and *Bph4*, resulting in the IR series of rice varieties, including IR26, IR36, IR54, and IR74. These resistant varieties show good initial effectiveness in controlling BPH infestation [117,118]. However, with the emergence of new biotypes of the BPH, the aforementioned resistant varieties have either lost their resistance or are gradually losing their resistance [119,120,121]. Practical experience has demonstrated that breeding resistant varieties, particularly those exhibiting broad-spectrum and durable resistance, is an effective approach for controlling BPH damage. However, breeding resistant varieties requires specialized and large-scale insect resistance screening, which many breeding institutions lack. With the continuous advancement and improvement of molecular breeding technologies, the use of molecular markers closely linked or co-segregating with resistance genes provides convenience for the direct selection of insect-resistant genes in breeding programs [122].

Furthermore, studies have shown that varieties with multiple genes controlling insect resistance exhibit stronger and broader-spectrum durable resistance compared to those controlled by a single gene. Currently, the mainly utilized resistance genes include *Bph1*, *Bph2*, *Bph6*, *Bph9*, *Bph12*, *Bph14*, *Bph15*, and *Bph18*. Breeders have employed marker-assisted selection (MAS) to combine these genes with the different disease- and insect-resistant genes, resulting in the development of a large number of resistant breeding materials and varieties with multiple resistance genes (Table 3). For example, the two-line hybrid rice variety Wei-Liang-You 7713 combines the broad-spectrum resistance genes *Bph6* and *Bph9*, while Hua-Liang-You 2171 and Yi-Liang-You 311 combine *Bph14* and *Bph15*. These resistant rice varieties have been widely promoted and applied in production, providing a solid foundation for the effective control of the BPH.

## 7. Discussion

So far, according to relevant literature reports, 70 BPH-resistant genes/QTLs (50 genes and 20 QTLs) have been successfully characterized in rice. However, there is a significant issue of gene/QTL duplication, with as many as 26 cases of duplication. For example, Yang et al. (2019) [65] have positioned *Bph37* between RM302 and YM35 in chromosome 1 in IR64, while Zhou et al. (2021) [66] has cloned a resistance gene on chromosome 6 in SE382 and also named it *Bph37*. Similarly, Tan et al. (2022) [69] have located the resistance gene *Bph41* between SWRm_01617 and SWRm_01522 in chromosome 4, and in the same year, Wang et al. (2022) [70] have reported *Bph41*, locating it between the molecular markers W4-4-3 and W1-6-3 on chromosome 4. As a result, the nomenclature of BPH-resistant genes has become confusing. In addition, some researchers do not follow the rice gene or QTL nomenclature rules based on recommendations from the Committee on Gene Symbolization, Nomenclature, and Linkage (CGSNL) of the Rice Genetics Cooperative, which also leads to confusion in the naming of BPH-resistant gene or QTL. This paper aimed to comprehensively review and summarize all reported BPH-resistant genes, with the goal of minimizing duplication and confusion, facilitating effective communication among researchers in the field, and promoting the rapid development of this discipline. To address the issue of confusing nomenclature for BPH-resistant genes, we propose that relevant research institutions, including the International Rice Research Institute, take prompt action to update and stay informed about new research advancements in resistance genes. Additionally, researchers should exercise caution when assigning names to newly discovered BPH-resistant genes, ensuring they are well informed about the latest relevant literature to avoid duplicating names already assigned to known genes while the naming of BPH resistance gene or QTL should follow the rice gene or QTL nomenclature rules recommended by CGSNL [141]. For example, *Bph28*(*t*) represents an uncloned BPH-resistance gene, the suffix “(t)” indicates ‘tentative’ locus designation; *qBph4.2* represents a resistance QTL.

Currently, although 70 BPH-resistant genes/QTLs (50 genes and 20 QTLs) have been identified, the application of these genes in BPH-resistant breeding is severely limited, resulting in a lack of BPH-resistant varieties in production. There are four main reasons for this limitation. Firstly, it is important to note that BPHs are migratory insects, and their mating across different biotypes from various regions can result in the emergence of new biotypes that possess the ability to overcome resistance genes. This phenomenon poses a significant challenge to the long-term effectiveness of resistant varieties. The rate at which resistance breeding occurs often fails to keep pace with the rapid variation of biotypes in BPHs, rendering previously resistant varieties ineffective within a few years after their initial promotion. Secondly, most resistance genes can only combat a specific local biotype of the BPH, lacking broad-spectrum and persistent resistance genes that can be widely applied in resistance breeding. Thirdly, BPH-resistant genes derived from different genetic backgrounds exhibit varying resistance effects under different genetic backgrounds and environmental conditions, resulting in suboptimal resistance performance in certain materials. Fourthly, most BPH-resistant genes are harbored in wild rice, and these genes are often closely linked to undesirable traits, making it challenging to separate them. Additionally, some wild rice species are incompatible with cultivated rice through hybridization, further complicating the transfer of target resistance genes to cultivated rice. Furthermore, insect-resistant breeding requires extensive insect resistance identification and screening, necessitating large-scale rearing and collection of insect sources for resistance evaluation. Unfortunately, most rice breeding institutions lack dedicated departments and teams for such research, which has also hindered the development of BPH-resistant breeding. However, with the increasing discovery and identification of excellent BPH-resistant gene resources and the maturation of molecular breeding technologies, China has made significant progress in BPH-resistant breeding. In recent years, Chinese rice breeders have combined conventional breeding methods with molecular breeding approaches to develop a series of new BPH-resistant rice varieties. These varieties have undergone provincial or national rice variety approval and have been extensively promoted in production. Examples include Luohong 4A and Yi Liangyou 311 (*Bph14* + *Bph15*), the world’s first BPH-resistant male sterile rice lines developed by Wuhan University; Weiliangyou 7713 (*Bph6* + *Bph9*) developed by Yuan Longping High-tech Agriculture Co., Ltd.; Huali 2171 (*Bph14* + *Bph15*) developed by Huazhong Agricultural University [125,142]; the high BPH-resistant fragrant rice variety Xinxiangzhan 1 and male sterile lines Guangdao A, Guangsixiang A, and Yinxian A developed by Guangxi University using broad-spectrum and persistent resistance source BPHR96. These varieties have been used to create three moderately resistant hybrid rice varieties: Guangdaoyou 3318, Guangsixiangyou 3216, and Yinxianyou Jinsizhan. These new BPH-resistant varieties have been successfully applied and promoted in production, establishing the genetic foundation for green control of BPHs and contributing significantly to ensuring food security.

The rapid development of molecular biology techniques has greatly enhanced our understanding of the functional aspects of BPH-resistant genes in rice. It has also provided increasingly clear insights into the genetic regulatory network and molecular mechanisms underlying rice resistance to BPH. This deeper understanding has deepened our knowledge of the interactions among rice, BPHs, and the environment, offering new perspectives for BPH-resistant breeding in rice. These advancements will drive the selection, breeding, and widespread application of BPH-resistant varieties, laying the foundation for sustainable and environmentally friendly control of the BPH.

## Figures and Tables

**Figure 1 ijms-24-12061-f001:**
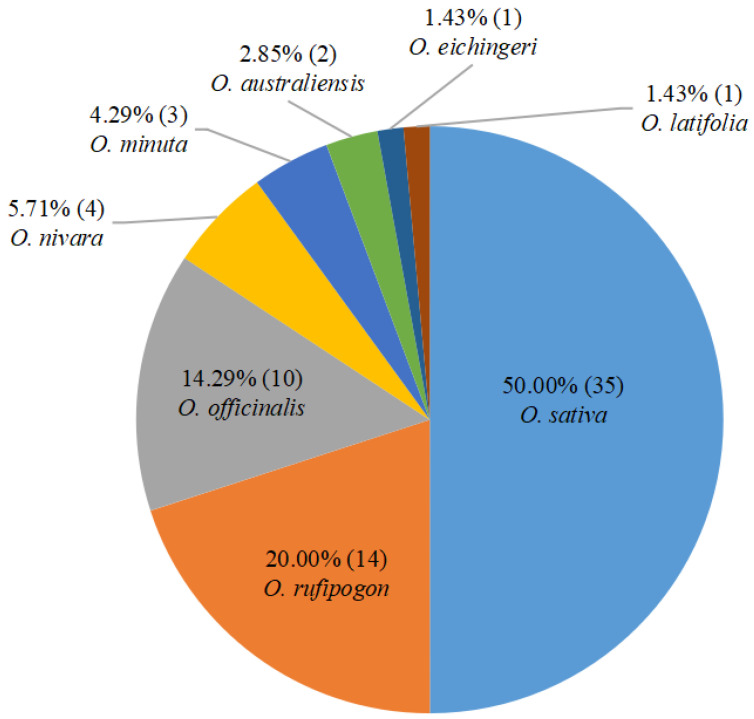
Distribution of 70 BPH-resistant genes/QTLs in rice germplasm resources. Numbers in parentheses indicate the number of resistance genes or QTLs.

**Figure 2 ijms-24-12061-f002:**
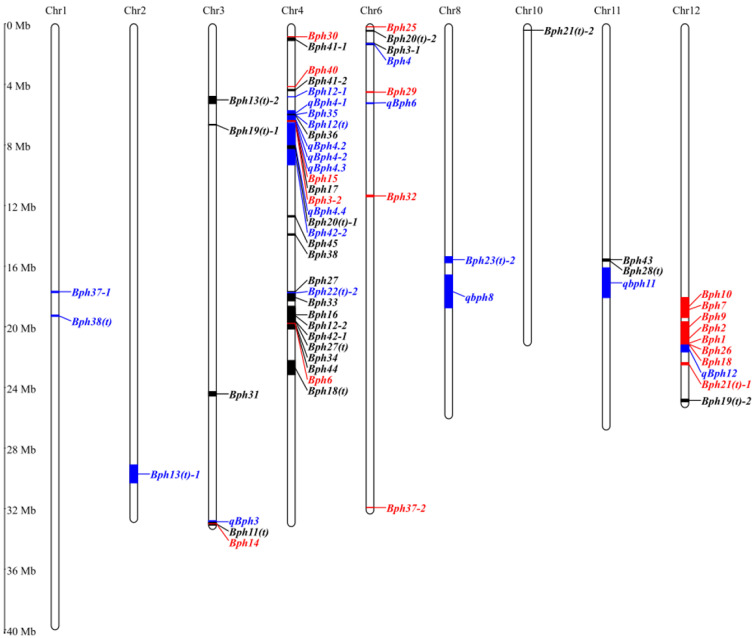
Chromosomal distribution of rice BPH-resistant genes/QTLs. A total of 70 genes/QTLs were identified in rice. Except for *Bph5* (*bph5*), *Bph8* (*bph8*), *Bph23*(*t*)*-1* [*Bph23*(*t*)], *Bph24*(*t*) [*bph24*(*t*)], *Bph39*(*t*) [*bph39*(*t*)], and *Bph40*(*t*)*-1* [*bph40*(*t*)], the remaining 64 genes/QTLs (44 genes and 20 QTLs) were mapped on chromosomes 1, 2, 3, 4, 6, 8, 10, 11, and 12, respectively. Black represents genes that have been located but not cloned. Red represents the cloned genes, while blue represents the QTLs.

**Figure 3 ijms-24-12061-f003:**
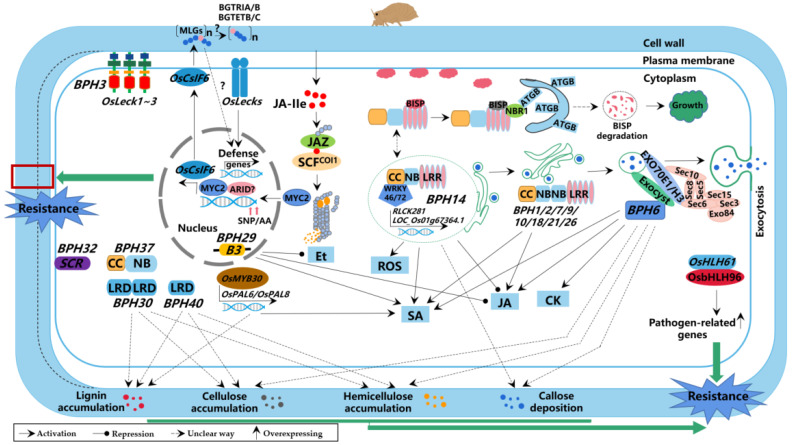
Resistance molecular mechanism of BPH-resistant genes in rice. BPH resistance genes regulation includes protein degradation, phytohormones, transcriptional regulatory factors, expression of defense genes, and other factors. These regulation pathways induce callose accumulation in sieve tubes, leading to sieve tube blockage, as well as cellulose, hemicellulose, and lignin accumulation in the cell wall of leaf sheaths, resulting in cell wall thickening, which inhibits the BPH feeding, growth, and reproduction, ultimately forming a molecular mechanism of resistance to brown planthopper invasion. The solid lines + arrow represent the activation way, the solid lines + endpoint represent the repression way, “?” and the dashed lines+arrow represent the unclear way. [Refering to the model map of BPH resistance molecular mechanism of Zheng et al. (2021) [92], and complementing and perfecting the molecular network mechanism related to all BPH-resistant genes].

**Table 1 ijms-24-12061-t001:** BPH-resistant genes/QTLs in rice.

Genes/QTLs	Chromosome	Germplasm	Linked Markers	Position (Mbp)	PEV	Reference
*Bph1*	12L	Mudgo	pBPH4-pBPH14	22.86	Major gene	[20,21]
*Bph2* (*bph2*)	12L	ASD7	RM463-RM7102	22.87~22.89	Major gene	[22,23]
*Bph3-1* (*Bph3*) #	6S	Ptb33, Rathu Heenati	RM589-RM588	1.38~1.48	Major gene	[24]
*Bph3-2* (*Bph3*) ***#	4S	Rathu Heenatis	RHD9-RHC10	6.20~6.97	Major gene	[25]
*Bph4* (*bph4*)	6S	Babawee	RM589-RM586	1.38~1.47	58.8–70.1%	[26,27]
*Bph5* (*bph5*)	—	ARC10550	—	—	Major gene	[28]
*Bph6 **	4L	Swarnalata	H-Y9	21.40	Major gene	[29]
*Bph7*	12L	T12	RM3448-RM313	19.95~20.87	38.3%	[30]
*Bph8* (*bph8*)	—	Chin Saba	—	—	Major gene	[31]
*Bph9 **	12L	Pokkali	InD2-RsaI	22.85~22.97	Major gene	[32,33]
*Bph10* (*bph10*)	12L	*O. australiensis*	RG457	19.55~26.98	Major gene	[34]
*Bph11*(*t*) (*bph11*)	3L	*O. officinalis*	G1318	35.60~35.80	Major gene	[35,36]
*Bph12-1* (*Bph12*) #	4S	B14 (*O. officinalis*)	RM16459-RM1305	5.21~5.56	73.8%	[37]
*Bph12-2* (*bph12*) #	4L	*O. officinalis*	G271-R93	20.34~21.31	Major gene	[38]
*Bph12*(*t*)	4S	*O. latifolia*	RM261-RM8213	4.44~6.57	70.6 %	[39]
*Bph13*(*t*)*-1* [*Bph13*(*t*)] #	2L	*O. eichingeri*	RM240-RM250	31.50~32.78	90%	
*Bph13*(*t*)*-2* [*Bph13*(*t*)] #	3S	IR54745-2-21 (*O. officinalis*)	RG100-RG191	5.18~5.70	Major gene	
*Bph14 **	3L	B5 (*O. officinalis*)	SM1-G1318	35.68~35.70	Major gene	[40]
*Bph15 **	4S	B5 (*O. officinalis*)	RG1-RG2	6.68~6.90	Major gene	[41]
*Bph16*	4L	*O. officinalis*	G271-R93	20.17~21.14	Major gene	[38]
*Bph17*	4S	Rathu Heenati	RM8213-RM5953	4.44~9.38	Major gene	[42]
*Bph18*(*t*) [*bph18*(*t*)]	4L	*O. rufipogon*	RM273-RM6506	24.05~25.05	Major gene	[43]
*Bph18* *	12L	IR65482-7-216-1-2 (*O. australiensis*)	BIM3-BN162	22.88	Major gene	[44,45]
*Bph19*(*t*)*-1* [*bph19*(*t*)] #	3S	AS20-1	RM6308-RM3134	7.18~7.24	Major gene	[46]
*Bph19*(*t*)*-2* [*bph19*(*t*)] #	12L	*O. rufipogon*	RM17	26.98	Major gene	[43]
*Bph20*(*t*)*-1* [*Bph20*(*t*)] #	4S	IR71033-121-15 (*O. minuta*)	B42-B44	8.76	Major gene	[47]
*Bph20*(*t*)*-2* [*bph20*(*t*)] #	6S	*O. rufipogon*	BYL7-BYL8	0.47~0.53	Major gene	[48]
*Bph21*(*t*)*-1* [*Bph21*(*t*)] #	12L	IR71033-121-15 (*O. minuta*)	S12094A-B122	24.20~24.36	Major gene	[47]
*Bph21*(*t*)*-2* [*bph21*(*t*)] #	10S	*O. rufipogon*	RM222-RM244	2.62~5.00	Major gene	[48]
*Bph22*(*t*)*-1*[*Bph22*(*t*)] #	4L	*O. glaberrima*	RM471-RM5742	18.99~21.56	Major gene	
*Bph23*(*t*)*-1* [*Bph23*(*t*)] #	—	*O. minuta*	—	—	Major gene	[49]
*Bph22*(*t*)*-2* [*bph22*(*t*)] #	4L	*O. rufipogon*	RM8212-RM261	19.11~19.57	11.3%	[50]
*Bph23*(*t*)*-2* [*bph23*(*t*)] #	8L	*O. rufipogon*	RM2655-RM3572	16.63~17.07	14.9%	[50]
*Bph24*(*t*) [*bph24*(*t*)]	—	IR73678-6-9-B (*O. rufipogon*)	—	—	Major gene	[49]
*Bph25*	6S	ADR52	S00310-RM8101	0.21	Major gene	[51]
*Bph26 **	12L	ADR52	DS72B-DS173B	22.87~22.89	Major gene	[52]
*Bph27*	4L	GX2183 (*O. rufipogon*)	RM16846-RM16888	19.12~19.50	Major gene	[53]
*Bph27*(*t*)	4L	Balamawee	Q52-Q20	20.79~21.33	Major gene	[54]
*Bph28*(*t*)	11L	DV85	Indel55-Indel66	16.90~16.96	Major gene	[55]
*Bph29* (*bph29*) ***	6S	RBPH54 (*O. rufipogon*)	BYL8-BID2	0.48~0.49	Major gene	[56]
*Bph30 **	4S	AC-1613	SSR28-SSR69	0.92~0.94	Major gene	[57,58]
*Bph31*	3L	CR2711-76	PA25-RM2334	26.25~26.57	Major gene	[59]
*Bph32 **	6S	Ptb33	RM19291-RM8072	12.23~12.36	Major gene	[60]
*Bph33*	4S	KOLAYAL, PPLIYAL	H99-H101	19.29~19.79	Major gene	[61]
*Bph34*	4L	IRGC104646 (*O. nivara*)	RM17007-RM1699	21.32~21.47	Major gene	[62]
*Bph35*	4S	RBPH660 (*O. rufipogon*)	RM3471-PSM20	6.28~6.94	51.27%	[63]
*Bph36*	4S	GX2183 (*O. rufipogon*)	S13-X48	6.46~6.50	Major gene	[64]
*Bph37-1* (*Bph37*) #	1L	IR64	RM302-YM35	19.10~19.20	36.9%	[65]
*Bph37-2* (*Bph37*) **#*	6S	SE382	—	3.45	Major gene	[66]
*Bph38*(*t*)	1L	Khazar	SNP693369-id 10112165	20.80~20.90	35.91%	[67]
*Bph38*	4L	GX2183 (*O. rufipogon*)	YM112-YM190	15.00~15.10	Major gene	[68]
*Bph40 **	4S	SE232, SE67, C334,	—	4.48~4.49	Major gene	[57]
*Bph41-1* (*Bph41*) #	4S	SWD10	SWRm_01617-SWRm_01522	0.90~1.10	Major gene	[69]
*Bph41-2* (*Bph41*) #	4S	GXU202 (*O. rufipogon*)	W4_4_3-W1_6_3	4.68~4.78	Major gene	[70]
*Bph42-1* (*Bph42*) #	4S	SWD10	SWRm_01695- SWRm_00328	20.60~21.80	Major gene	[69]
*Bph42-2* (*bph42*) #	4S	*O. rufipogon*	RM16282-RM16335	9.07~9.58	29%	[71]
*Bph43*	11S	IRGC 8678	InDel16_22-InDel16-30	16.79~16.90	Major gene	[72]
*Bph44*	4L	Balamawee	Q31-RM17007	21.38~21.47	Major gene	[73]
*Bph45*	4L	Tainung71 (*O. nivara*)	—	13.70~13.80	Major gene	[74]
*qBph3*	3L	IR02W101 (*O. officinalis*)	t6-f3	35.63~35.47	28%	[75]
*qBph6*	6S	IR71033-121-15	RM469-RM568	5.64~5.71	19.6%	[76]
*qbph8*	8L	Swarnalata	RM339-RM515	17.94~20.28	6.6%	[77]
*qbph11*	11L	DV85	XNpb202-C1172	17.43~19.56	68.4%	[78]
*qBph4-1* (*qBph4*) #	4S	IR02W101 (*O. officinalis*)	P17-xc4_27	6.70~6.90	35%	[75]
*qBph4-2* (*qBph4*) #	4S	BP360e	RM16382-INDEL4-5	6.20~6.70	56%	[79]
*qBph4.2*	4S	IR65482-17-511 (*O. officinalis*)	RM261-XC4-27	6.58~6.89	36–44%	[80]
*qBph4.3*	4S	*Salkathi*	RM551-RM335	0.177~0.688	37.02%	[81]
*qBph4.4*	4S	*Salkathi*	RM335-RM5633	0.688~13.07	7.1%	[81]
*qBph12*	12L	ASD7	RM28466-RM7376	22.94~23.44	28.8%	[76]
*Bph39*(*t*) [*bph39*(*t*)]	—	*O. nivara*	—	—	Major gene	[82]
*Bph40*(*t*)*-1* [*bph40*(*t*)]	—	*O. nivara*	—	—	Major gene	[82]

L stands for the long arm; S stands for the short arm; “*” means it has been cloned; the original name in brackets; “#” means the gene with a repeated name; PEV: Phenotypic variation.

**Table 2 ijms-24-12061-t002:** The cloned BPH-resistant genes in rice.

Gene	Chr.	Donor Parent	Subcellular Localization	Encoded Protein	Year	Reference
*Bph1*	12L	Mudgo	Endomembrane system	CC-NB-NB-LRR	2016	[33]
*Bph2*	12L	ASD7	Endomembrane system	CC-NB-NB-LRR	2016	[33,52]
*Bph3*	4S	Rathu Heenati	Plasma membrane	LRK	2015	[25]
*Bph6*	4L	Swarnalata	Exocyst	LRD	2018	[29]
*Bph7*	12L	T12	Endomembrane system	CC-NB-NB-LRR	2016	[33]
*Bph9*	12L	Pokkali	Endomembrane system	CC-NB-NB-LRR	2016	[33]
*Bph10*	12L	IR65482-4-136-2-2	Endomembrane system	CC-NB-NB-LRR	2016	[33]
*Bph14*	3L	B5	Nucleus andcytoplasm	CC-NB-LRR	2009	[40]
*Bph15*	4S	B5	Plasma membrane	LRK	2013	[83]
*Bph18*	12L	IR65482-7-216-1-2(*O. australiensis*)	Endomembrane system	CC-NB-NB-LRR	2016	[44]
*Bph21*	12L	IR71033-121-15 (*O. minuta*)	Endomembrane system	CC-NB-NB-LRR	2016	[33]
*Bph26*	6S	ADR52	Endomembrane system	CC-NB-NB-LRR	2014	[33,52]
*Bph29*	6S	RBPH54	Nucleus	B3 DNA	2015	[56]
*Bph30*	4S	AC-1613	Endomembrane system	LRD	2021	[57]
*Bph32*	6S	Ptb33	Plasma membrane	SCR	2016	[60]
*Bph37*	6L	SE382	—	CC-NB	2021	[66]
*Bph40*	4S	SE232, SE67, C334	—	LRD	2021	[57]

Chr.: chromosome; CC-NB-LRR: coiled coil-nucleotide binding-leucine rich repeat: LRD, leucine-rich repeat domain; LRK: lectin receptor kinase; SCR: short consensus repeat.

**Table 3 ijms-24-12061-t003:** List of BPH resistance genes pyramided in rice.

Pyramided Genes/QTLs	Trait	Resistant Lines/Varieties	References
*Bph1* + *Bph2*	BPH resistance	Tsukushibare	[123]
*Bph6* + *Bph9*	BPH resistance	9311	[124]
*Bph6* + *Bph9*	BPH resistance	Wei-Liang-You 7713	[125]
*Bph6* + *Bph12*	BPH resistance	93-11, Nipponbare	[37]
*Bph3* + *Bph27*	BPH resistance	93-11, Ningjing3	[126]
*Bph14* + *Bph15*	BPH resistance	Minghui 63; Huahui938; Huang-Hua-Zhan; Hua-Liang-You2171; Yi-Liang-You 311	[125,127,128,129,130]
*Bph3 + Bph14 + Bph15*	BPH resistance	Hemeizhan	[131]
*Bph14 + Bph15 + Bph18*	BPH resistance	9311	[132]
*Bph14 + Bph15 +Xa21*	BPH and bacterial blight resistance	Yuehui9113	[133]
*Bph3 + Pita + Pi1 + Pi2 + xa5 + Xa23*	BPH, bacterial blight and blast resistance	HN88	[134]
*Qbph12 + Sub1*	BPH resistance and submergence tolerance	KDML105	[135]
*Bph14 + Bph15 + Stv-b^i^*	BPH and rice stripe disease resistance	Shengdao15, Shengdao16, Xudao3	[136]
*Bph6 + Bph9 + Gn8.1 + Rf3 + Rf4 + Rf5 + Rf6*	BPH resistance, big panicle and fertility restoration gene	9311	[137]
*Bph18 + Xa40 + qSTV11^SG^ + Pib + Pik*	BPH resistance, bacterial blight, rice stripe virus, and blast	Junam	[138]
*Bph25 + Bph26*	BPH resistance	Taichung 65	[51]
*Bph17 + Bph21 + Bph32*	BPH resistance	IR24	[139]
*Bph3* + *Bph17* + *Pi9 + Xa4 + xa5* + *Xa21* + *Gm4* + *Gm8* + *qDTY_1.1_* + *qDTY_3.1_*	BPH resistance, gall midge, blast, bacterial leaf blight and drought tolerance	Swarna + drought	[140]

## Data Availability

Not applicable.

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
