# Peer review of "Recent Advances in Molecular Mechanism and Breeding Utilization of Brown Planthopper Resistance Genes in Rice: An Integrated Review"

_ijms, 2023, doi:10.3390/ijms241512061_

Round 1

Reviewer 1 Report

good work 

Author Response

Dear reviewer,

We are very grateful to you for your tremendous time and effort in reviewing our paper, as well as for your positive comments of our manuscript. Please see the attachment.

Reviewer 2 Report

Yan et manuscript “Recent Advances in molecular ……….……. An Integrated review” attempted to compile literature related to genetic and molecular pathways involved in controlling Brown Planthopper in rice. They are also using/proposing nomenclature to unify known QTLs.

I have some serious concerns/comments:

1. They need to clarify and differentiate QTL and genes in entire manuscript. Current version is interchanging both terms. Initial mapping is always QTL. If there is further fine mapping and gene is identified and cloned. Lines 82: “To date, a total of 70 BPH-resistant genes/QTL have been identified…..”  is not a correct way.  One QTL can have one gene or many genes within each QTL.  For section 3, it need to say map QTLs.

2. Section 3 they wrote QTL name (t). What is t?

3. Table 1. Add year columns when each QTL were identified.

4. Table 1. QTL column need more space to make each line fit into one row. I think it needs align left

5. Table 1 . it will be good to have how effect of QTL on phenotype. That is how much phenotype is explained by QTL.

6. Each figure needs more detailed information in legend

7. Section 4. When they clone, does it mean gene in mapped QTL is identified and validated its effect. Table 2 is confusing. How come Bph1, Bph7,9,10,21 are cloned in one publication. Reference # 33. Usually it take years to clone one gene and it comes with one original publications. That makes me concern that if authors has clear idea on how to mention QTL and gene cloned. If there are really 17 genes have been cloned from their respective mapped QTL, I expect 17 original publications.

8. Line 116. Bph3 (also known as Bph17) ???. Those are listed as separate in Table 1

9. Figure 3. I assume all the BPH gene whose functions has been characterized are shown here. Also, any BPH gene in figure need to be explained well in figure legend

10. Is Figure 3 is redrawn from Zheng et l 2021. If yes, did authors take permission

11. Seems like much of final conclusion towards highlighting China achievements in developing resistant lines (Table 3). Very surprising no other lines from other part of the world?

12. Figure 1: “Indica” should be written as “O. sativa

No major issue found

Author Response

Dear reviewer,

I am very grateful to your comments for the manuscript. According with your advice,we amended the relevant part in manuscript. Some of your questions were answered be low. Please see the attachment.

Response to Reviewer 2 Comments

Point 1: They need to clarify and differentiate QTL and genes in entire manuscript. Current version is interchanging both terms. Initial mapping is always QTL. If there is further fine mapping and gene is identified and cloned. Lines 82: “To date, a total of 70 BPH-resistant genes/QTL have been identified…..”  is not a correct way. One QTL can have one gene or many genes within each QTL.  For section 3, it needs to say map QTLs.

Response 1: As suggested by the reviewer, we have clarified and differentiated QTL and genes throughout the paper. Firstly, regarding the genetic analysis of brown planthopper resistance gene or QTL, if the segregation ratio of resistant and susceptible plants in the F2 genetic population fits to the Mendelian law of genetic segregation or independent distribution, then BPH resistance of the variety (line) is controlled by major gene [e.g. Bph28(t)] (Reference 55), and if resistance value of the single plant in the F2 genetic population conforms to normal distribution, then BPH resistance of the variety (line) is controlled by QTL (e.g. qBph4.2) (Reference 80), therefore, BPH resistance may be either controlled by gene or by QTL in different varieties (lines). Secondly, regardless of the initial or fine localization of gene or QTL, they can only be mapped in a certain interval on a chromosome, but ultimately there is only one gene associated with BPH resistance. Thus, a total of 50 resistance genes and 20 QTLs were identified in this paper.

Point 2: Section 3 they wrote QTL name (t). What is t?

Response 2: The current rules for gene names in rice are based on recommendations from the Committee on Gene Symbolization, Nomenclature and Linkage (CGSNL) of the Rice Genetics Cooperative, rice Gene name with a "t" indicates ‘tentative’ locus designation (e. g. Bph28(t)) (Reference 55). Rice QTL nomenclature rules indicate that each QTL name should be italicized and start with a lowercase letter “q” to indicate that it is a QTL, followed by a two to five letter standardized “trait name” (e. g. BPH for Brown planthopper), a number designating the rice chromosome on which it occurs (1–12), a period (“.”), and a unique identifier to differentiate individual QTLs for the same trait that reside on the same chromosome (e. g. qBph4.2) (Reference 80). However, there are also a number of researchers who do not follow the naming rules for gene or QTL in rice, thus leading to confusion in the naming of gene or QTL.

Point 3: Table 1. Add year columns when each QTL were identified.

Response 3: The identification and localization of a BPH resistance genes/QTL is a study with a very large amount of work, and it is not easy to clone a gene/QTL, so it is possible that different research progresses on the same resistance gene/QTL may be published several times in different professional journals, so we considered the time to complete the cloning of a gene/QTL was more meaningful, and thus we added the time to complete the cloning of a BPH resistance genes/QTL to the 6th column of Table 2.

Point 4: Table 1. QTL column need more space to make each line fit into one row. I think it needs align left.

Response 4: As suggested by the reviewer, we have adjusted Table 1 so that each row has enough space and left-aligned the contents of the table.

Point 5: Table 1. it will be good to have how effect of QTL on phenotype. That is how much phenotype is explained by QTL.

Response 5: As suggested by the reviewer, we have added QTL effect values to explain phenotypic variation to the 6th column of Table 1.

Point 6: Each figure needs more detailed information in legend.

Response 6: We have added explanations to the legend of each figure. For example, Figure 1 adds explanations of the numbers in parentheses, Figure 2 adds different colors to indicate resistance genes that have been localized or cloned as well as QTLs, and Figure 3 adds different pathways represented by different arrows.

Point 7: Section 4. When they clone, does it mean gene in mapped QTL is identified and validated its effect. Table 2 is confusing. How come Bph1, Bph7,9,10,21 are cloned in one publication. Reference # 33. Usually it take years to clone one gene and it comes with one original publications. That makes me concern that if authors have clear idea on how to mention QTL and gene cloned. If there are really 17 genes have been cloned from their respective mapped QTL, I expect 17 original publications.

Response 7: To date, a total of 17 genes associated with resistance to BPH in rice have been cloned, but of which eight BPH-resistance genes (Bph1, Bph2, Bph7, Bph9, Bph10, Bph18, Bph21, and Bph26) have been reported on chromosome 12L (Reference 20 and 21, 22 and 23, 30, 32 and 33, 34, 44 and 45, 47, 52). Among them, Bph2 is identical to Bph26 (Reference 52), Bph9 shares 91.32% and 96.05% nucleotide sequence identity with Bph26 and Bph18, respectively (Reference # 33). Together with their chromosome position and genomic sequence, these data confirmed that Bph9, Bph2/26, and Bph18 are alleles (Reference 33). To test whether the remaining four BPH-resistance genes (Bph1, Bph7, Bph10, and Bph21) in this cluster are also alleles of Bph9/26, Zhao et al. (2016) cloned the full-length cDNAs of the gene corresponding to Bph9 were obtained from the varieties with which these four genes were originally mapped, and then they found that the nucleotide and deduced amino acid sequences in rice varieties harboring Bph1, Bph10, and Bph21 were identical to Bph18, whereas the sequence from T12 (harboring Bph7) indicated a distinct allele. Thus, their findings demonstrated that the eight genes in this cluster are multiple alleles of the same locus. So, the cloning of Bph1, Bph7, Bph9, Bph10, Bph21 genes were all reported from the same paper (Reference 33). 

Point 8: Line 116. Bph3 (also known as Bph17) ???. Those are listed as separate in Table 1.

Response 8: We are very sorry that we refer to the literature of other researchers, and such a statement is not rigorous. Although both Bph3 and Bph17 were located on chromosome 4 in Rathu Heenati, but their localization intervals were 6.20 Mb - 6.97 Mb and 4.44 Mb - 9.38 Mb, respectively, and there is no evidence that they are the same gene. Therefore, they should be listed separately in Table 1.

Point 9: Figure 3. I assume all the BPH gene whose functions has been characterized are shown here. Also, any BPH gene in figure need to be explained well in figure legend.

Response 9: Figure 3 involves a complex molecular regulatory network of 17 BPH resistance genes (Bph1/2/7/9/10/18/21/26, Bph3, Bph6, Bph14, Bph15, Bph29, Bph30, Bph32, Bph37, Bph40, where Bph1/2/7/9/10/18/21/26 are multiple alleles at the same locus) and five transcription factors (OsHLH61, OsbHLH96, OsCsIF6, and OsMYB30), which activate a series of defense related signaling pathways, including protein degradation, phytohormones, transcriptional regulatory factors, expression of defense genes and other factors. These pathways induce callose accumulation in sieve tubes, leading to sieve tube blockage, as well as cellulose, hemicellulose, and lignin accumulation in the cell wall of leaf sheaths, resulting in cell wall thickening, which inhibits the brown planthopper feeding, growth, and reproduction, ultimately forming a molecular mechanism of resistance to brown planthopper invasion.

Point 10: Is Figure 3 is redrawn from Zheng et al. 2021. If yes, did authors take permission.

Response 10: Zheng et al. (2021) described the molecular regulatory networks of the BPH resistance genes Bph6, Bph1/2/7/9/10/18/21/26, Bph14, Bph15, and Bph30, while Figure 3 in our paper referred to their model map of the BPH resistance molecular mechanisms, and added the molecular regulatory networks of Bph3, Bph29, Bph32, Bph37, Bph40, OsHLH61,OsbHLH96, OsCsIF6 and OsMYB30 against BPH in rice, as well as a new mechanism for fine regulation of resistance-growth balance by BISP (BPH14-interacting salivary protein)-BPH14-OsNBR1(Selective autophagy cargo receptor) interaction system. we systematically refine the molecular network mechanisms associated with all brown planthopper resistance genes in Figure 3 of the manuscript.

Point 11: Seems like much of final conclusion towards highlighting China achievements in developing resistant lines (Table 3). Very surprising no other lines from other part of the world?

Response 11: We apologized for not being able to fully summarize the papers related to the development of BPH resistant lines. Brown planthopper occurs mainly in Asia, and China is one of the countries with the most serious brown planthopper infestation, so China has a very large number of people worked on research related to BPH and a relatively large number of papers published. The researchers engaged in studies related to BPH in other Asian countries are also very good. In Table 3, we have added three papers on BPH resistant varieties (lines) from other Asian countries that we have found in the NCBI database.

Point 12: Figure 1: “Indica” should be written as “O. sativa”.

Response 12: Thanks to the reviewer for the reminder. We have revised "Indica" to "O. sativa" in Figure 1 (line 82).

Reviewer 3 Report

Manuscript entitled “Recent Advances in Molecular Mechanism and Breeding Utilization of Brown Planthopper Resistance Genes in Rice: An Integrated Review” by Liuhui Yan et al. is a good revision of the resistance to Brown Planthopper in rice.

The work is interesting but, in my opinion, the manuscript presents some issues mainly in the Figures,Tables and References and, as well as other things, that need to be checked to make this work suitable for publication.

1. Main issues:

1.1. Regarding Figures.

Line 73. Please correct “Asia.” with “Asia (Figure 1).”

Figure 1. Is the Oryza sativa lost?

Line 84. Please correct “Fig.2” with “Figure 2.”. Review the entire document

Figure 2. Only 58 of the 70 resistant genes/quantitative 82 trait loci (QTLs) are represented in the figure.

1.2. Regarding Tables.

Table 1. review the entire format (e.g. Line 34 Please correct “O.australiensis” with “O. australiensis”, Line 37 Please correct “B14(O. officinalis)” with “B14 (O. officinalis)”,  etc.

Table 1. Line 68. O. rufipogon should be in italics

Table 2. Line 44. Please correct “O.austrliensis” with “O. austrliensis

Tables 1 and 2. Please check “austrliensis”

1.3. Regarding section References

The references list has numerous format mistakes. According to the Instructions for Authors section of this journal: Author 1, A.B.; Author 2, C.D. Title of the article. Abbreviated Journal Name Year, Volume, page range.

Please put the volume in italics.

Revise the italics (e.g. Line 480 Pleas correct “Oryza sativa” with “Oryza sativa”, revise the entire manuscript).

The DOI of each reference must be removed.

Almost all the words in the title begin with capital letters. Many of them should not be in that format (reference 22, etc.). Please revise all references.

The magazine's abbreviated name is dotted (e.g. Pestic. Biochem. Phys.) (line 499 please correct “Sci Rep” with “Sci. Rep.”). Please revise all references.

Line 486. The Journal Name is not Abbreviated. Please revise all references.

Reference 43. Are missing the volume and page range. Please revise all references.

Etc.

2. Additional comments

The cited reference should be separated from the last word. Please correct throughout the manuscript (e.g. Line 41 “population[1-3] ”).

Please check comma in “[4,5]”, is a comma with a strange format. Review the entire document (e.g. Lines 48, 56, 60, 337, etc.)

Lines 94 and 95. Please correct “Bph37-1, Bph38(t)-2, Bph13(t)-1, Bph23(t)-2, Bph28(t), and Bph43 are located on chromosomes 1, 2, 8, and 11, respectively” with “Bph37-1 and Bph38(t), Bph13(t)-1, Bph23(t)-2, Bph28(t) and Bph43 are located on chromosomes 1, 2, 8, and 11, respectively”.

Line 116. Bph3 is Bph3-2? Since you indicate a new nomenclature you should use it in the text

 Line 332. Please check Ttranscription

Line 397. Please correct “[65]has” with “[65] has”

Line 399. Please correct “[66]has” with “[66] has”

Line 414. The 70 genes have not been cloned

Lines 462-465. Chi Liu is missing.

Author Response

Dear reviewer,

Thank you for your positive comments and valuable suggestions to improve the quality of our manuscript. The main corrections in the paper and the responds to your comments are as flowing. Please see the attachment.

Response to Reviewer 3 Comments

Regarding Figures.

Point 1: Line 73. Please correct “Asia.” with “Asia (Figure 1).”

Response 1: Thanks to the reviewer for the reminder, but we thought it would be more appropriate to revise "Asia" to "Asia [18]" in line 74, and revised "cultivated rice" to "O. sativa (Figure 1, Table 1) " in line 75-76.

Point 2: Figure 1. Is the Oryza sativa lost?

Response 2: We have misrepresented the text and have revised "Indica" to "O. sativa" in Figure 1 (line 82).

Point 3: Line 84. Please correct “Fig.2” with “Figure 2.” Review the entire document

Response 3: We have corrected “Fig.2” with “Figure 2” (line 90) and checked the whole text. 

Point 4: Figure 2. Only 58 of the 70 resistant genes/quantitative trait loci (QTLs) are represented in the figure.

Response 4: To date, a total of 70 genes/QTLs were identified in rice. Except for Bph5 (bph5), Bph8 (bph8), Bph23(t)-1 [Bph23(t)], Bph24(t) [bph24(t)], Bph39(t) [bph39(t)], and Bph40(t)-1 [bph40(t)], the remaining 64 genes/QTLs (44 genes and 20 QTLs) were mapped on chromosomes 1, 2, 3, 4, 6, 8, 10, 11, and 12, respectively. Thus, there are only 64 genes/QTLs in Figure 2.

Regarding Tables.

Point 5: Table 1. review the entire format (e.g. Line 34 Please correct “O.australiensis” with “O. australiensis”, Line 37 Please correct “B14(O. officinalis)” with “B14 (O. officinalis)”,  etc.

Response 5: We have checked the full format of the Latin names of the various different wild rice species in Table 1, and have corrected “O.australiensis” and “B14(O. officinalis)”with “O. australiensis” and “B14 (O. officinalis) ” in Reference 34 and Reference 37of Table 1, respectively.

Point 6: Table 1. Line 68. O. rufipogon should be in italics

Response 6: Thanks to the reviewer for the reminder, we have modified O. rufipogon to be italicized in Reference 68 of Table 1.

Point 7: Table 2. Line 44. Please correct “O.australiensis” with “O. australiensis

Response 7: Thank you for pointing this out, we have corrected “O.austrliensis” with “O. austrliensis” in Reference 44 of Table 2.  

Point 8: Tables 1 and 2. Please check “austrliensis”

Response 8: Thanks for your careful checks, we have corrected “austrliensis” with “australiensis” in Tables 1 and 2 (Reference 44, 45).

Regarding section References

Point 9: The references list has numerous format mistakes. According to the Instructions for Authors section of this journal: Author 1, A.B.; Author 2, C.D. Title of the article. Abbreviated Journal Name Year, Volume, page range. Please put the volume in italics.

Response 9: Thank you for pointing this out, we have checked and corrected the formatting of references throughout the text and have italicized "Volume" in all references.

Point 10: Revise the italics (e. g. Line 480 Pleas correct “Oryza sativa” with “Oryza sativa”, revise the entire manuscript).

Response 10: Thank you for pointing this out, we have checked the Latin names for the entire manuscript and have revised them all to italics (e. g. line 520 correct “Oryza sativa” with “Oryza sativa”).

Point 11: The DOI of each reference must be removed.

Response 11: Thank you for pointing this out, we have removed the DOI from all references.

Point 12: Almost all the words in the title begin with capital letters. Many of them should not be in that format (reference 22, etc.). Please revise all references.

Response 12: We have changed the capital letters at the beginning of all words in the titles in all references to lower case letters as required by the “International journal of molecular science” journal for references.

Point 13: The magazine's abbreviated name is dotted (e. g. Pestic. Biochem. Phys.) (line 499 please correct “Sci Rep” with “Sci. Rep.”). Please revise all references.

Response 13: Thanks to the reviewer for the reminder. We have added dots to the end of the journal name abbreviations in all references (line 538 correct “Sci Rep” with “Sci. Rep.”).

Point 14: Line 486. The Journal Name is not Abbreviated. Please revise all references.

Response 14: We have abbreviated the journal names of all references in the format of the “International journal of molecular science” journal.

Point 15: Reference 43. Are missing the volume and page range. Please revise all references.

Response 15: We were really sorry for our careless mistakes. We have filled in the missing the volume and page range in Reference 43, and checked and corrected all references in the paper.

Additional comments

Point 16: The cited reference should be separated from the last word. Please correct throughout the manuscript (e. g. Line 41 “population[1-3] ”).

Response 16: Thanks to the reviewer for the reminder. We have separated the cited reference from the last word throughout the manuscript (e. g. line 42 “population[1-3] ”correct with “population [1-3]”).

Point 17: Please check comma in “[4,5]”, is a comma with a strange format. Review the entire document (e. g. Lines 48, 56, 60, 337, etc.)

Response 17: Thanks to the reviewer for the reminder. We have changed the formatting of commas in square brackets in cited references in the entire document (e. g. line 44 correct “[4,5]” with “[4,5]”).

Point 18: Lines 94 and 95. Please correct “Bph37-1, Bph38(t)-2, Bph13(t)-1, Bph23(t)-2, Bph28(t), and Bph43 are located on chromosomes 1, 2, 8, and 11, respectively” with “Bph37-1 and Bph38(t), Bph13(t)-1, Bph23(t)-2, Bph28(t) and Bph43 are located on chromosomes 1, 2, 8, and 11, respectively”.

Response 18: We sincerely thank the reviewer for careful reading. According to Figure 2 and Table 2, we missed three genes/QTL on chromosomes 8, 10 and 11 [qbph8, Bph21(t)-2, qbph11] and now need to add these three genes to the paper. Therefore “Bph37-1, Bph38(t)-2, Bph13(t)-1, Bph23(t)-2, Bph28(t), and Bph43 are located on chromosomes 1, 2, 8, and 11, respectively” should be revised to " Bph37-1, Bph38(t), Bph13(t)-1, Bph23(t)-2, qbph8, Bph21(t)-2, Bph28(t), Bph43 and qbph11 are located on chromosomes 1, 2, 8, 10 and 11, respectively" in lines 107-109.

Point 19: Line 116. Bph3 is Bph3-2? Since you indicate a new nomenclature you should use it in the text.

Response 19: We have differentiated the Bph3 genes located on chromosomes 6 and 4 using Bph3-1 and Bph3-2 in Figure 2, but Bph3 (Bph3-2) has been cloned and is known for its broad-spectrum resistance. It is already well known to researchers and is being widely used in BPH resistance breeding. Meanwhile, in order to respect the priority of the literature, we do not suitably use Bph3-2 in place of Bph3 in the full text.

Point 20: Line 332. Please check Ttranscription.

Response 20: We feel sorry for our carelessness. The “Ttranscription” has been corrected with “Transcription” in line 364.

Point 21: Line 397. Please correct “[65]has” with “[65] has”.

Response 21: Thank you for pointing this out. We have corrected “[65]has” with “[65] has” in line 439.

Point 22: Line 399. Please correct “[66]has” with “[66] has”.

Response 22: Thanks to the reviewer for the reminder. We have corrected “[66]has” with “[66] has” in line 440.

Point 23: Line 414. The 70 genes have not been cloned.

Response 23: We sincerely thank the reviewer for careful reading. We have revised “although 70 BPH-resistant genes have been identified and cloned” to “although 70 BPH-resistant genes/QTLs have been identified in rice” in line 455-456.

Point 24: Lines 462-465. Chi Liu is missing.

Response 24: Thanks for your careful checks. We have added "C.L." after "Z.M." and before "Y.Q." in line 504.

Round 2

Reviewer 2 Report

Author made good improvement from original version of the manuscript.

Response 2 about the current nomenclature should in the manuscript text as well

To avoid any confusions to the reader, authors need to include response 7 in the manuscript text as well.

Thanks

Author Response

Thank you very much for your the very good suggestions and comments on our manuscript, so that our paper can be improved. We added response 2 to lines 472-475, and added response 7 to lines 196-205 of the manuscript.